# Towards the Dependence on Parameters for the Solution of the Thermostatted Kinetic Framework

**Bruno Carbonaro** † **and Marco Menale** *,†

Dipartimento di Matematica e Fisica, Università degli Studi della Campania "L. Vanvitelli", Viale Lincoln 5, I-81100 Caserta, Italy; bruno.carbonaro@unicampania.it
* Correspondence: marco.menale@unicampania.it
† These authors contributed equally to this work.

**Abstract:** A complex system is a system involving particles whose pairwise interactions cannot be composed in the same way as in classical Mechanics, i.e., the result of interaction of each particle with all the remaining ones cannot be expressed as a sum of its interactions with each of them (we cannot even know the functional dependence of the total interaction on the single interactions). Moreover, in view of the wide range of its applications to biologic, social, and economic problems, the variables describing the state of the system (i.e., the states of all of its particles) are not always (only) the usual mechanical variables (position and velocity), but (also) many additional variables describing e.g., health, wealth, social condition, social rôle . . . , and so on. Thus, in order to achieve a mathematical description of the problems of everyday's life of any human society, either at a microscopic or at a macroscpoic scale, a new mathematical theory (or, more precisely, a scheme of mathematical models), called KTAP, has been devised, which provides an equation which is a generalized version of the Boltzmann equation, to describe in terms of probability distributions the evolution of a non-mechanical complex system. In connection with applications, the classical problems about existence, uniqueness, continuous dependence, and stability of its solutions turn out to be particularly relevant. As far as we are aware, however, the problem of continuous dependence and stability of solutions with respect to perturbations of the parameters expressing the interaction rates of particles and the transition probability densities (see Section The Basic Equations has not been tackled yet). Accordingly, the present paper aims to give some initial results concerning these two basic problems. In particular, Theorem 2 reveals to be stable with respect to small perturbations of parameters, and, as far as instability of solutions with respect to perturbations of parameters is concerned, Theorem 3 shows that solutions are unstable with respect to "large" perturbations of interaction rates; these hints are illustrated by numerical simulations that point out how much solutions corresponding to different values of parameters stay away from each other as $t \to +\infty$.

**Keywords:** kinetic theory; complex systems; stability; parameters; differential equations

**MSC:** 82B40; 37F05; 45M10; 35B30; 34A12

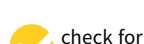

## 1. Introduction

The present paper deals with the system of equations governing the behavior of so-called *complex systems* (see Section 2 for details). Roughly speaking, a *complex system* is a set of a large number of individuals (particles) whose behavior is strongly influenced by their mutual interactions, in addition to external forces and possibly to a *thermostat* [1–4], so that the evolution of the system cannot be by no means *deterministic*, but must be described in terms of the *probability distribution fuction* on the set of possible values of a suitable variable describing the state of each individual. In this connection, it must be carefully noted that—though the notion of a complex system was originated in a purely mechanical framework and could be traced back to Boltzmann's Kinetic Theory of Gases [5–8]—a complex system is not nowadays considered as simply consisting of material particles, whose state is

completely described by the two variables *position* and *velocity*. The notion of complex system has been exported in several different contexts (biology [9,10], medicine [11–15], economy [16,17], psychology [18,19], social dynamics [20–23] . . . ), in which the state variables are non-mechanical and, in at least one case, vectorial [24].

Though the behavior of each particle of the system is of course deterministic, i.e., it is uniquely determined by its interactions with other particles; nevertheless, the number of particles and interactions is so large as to prevent us from following the evolution of the state of each particle. Accordingly, the model is based on the choice to describe the evolution of the system as a whole, turning the attention to the probability distribution on the states of particles; thus, the evolution equation takes the form (1) (see Section 2). As usual for nonlinear differential and integral equations, also in this case we have to tackle the classical problems about existence, uniqueness, continuous dependence, and stability with respect to initial values (and boundary values, when required). These problems have been tackled in [25–30].

In many cases of interest, the solutions to the problems about stability and continuous dependence of solutions depend on the coefficients of the equations, especially in the cases in which they are not constants but functions of the independent variables. In this last case, the question of whether two solutions, corresponding to the same assigned data but to two different systems of coefficients, are close when such are the coefficients spontaneously arises. This question seems to be of special relevance for Equation (1). As we shall see in more detail in Section 2, in Equation (1), denoting by $u$ the state variable and by $D_u$ the state space, that is the set of all possible values of $u$, we find two kinds of coefficients:

1.   the coefficient $\eta(u_*, u^*)$, a function defined on $D_u^2$, expressing the interaction rate of the particles whose state is $u_*$ with the particles whose state is $u^*$, i.e.,— roughly speaking—the number of their interactions per unit time;
2.   the coefficient $\mathcal{A}(u_*, u^*, u)$, a function defined on $D_u^3$, expressing the transition probability, i.e., the probability (density) that any individual in the state $u_*$, when interacting with a particle in the state $u^*$, *falls* in the state $u$.

In any context, it is obvious that, to different prescriptions on the form of function $\eta(u_*, u^*)$ or of function $\mathcal{A}(u_*, u^*, u)$, there will correspond different probability distributions on $D_u$ (or, in a strictly statistical interpretation, different distributions of relative frequencies on $D_u$ over the system). However, what should we expect about the dependence of the difference of distributions on the difference between prescribed coefficients? Should an accordingly small difference between the corresponding distributions correspond to *small* perturbations to the interaction rate or to the transition distribution?

These questions are quite similar to those posed in all the classical problems associated with differential and integro-differential equations, but—as far as we are aware—have not been tackled yet for Equation (1). Nevertheless, in view of the large number of applications of complex systems (and of Equation (1), which describes their evolution) to so many basic problems of collective life of the whole mankind (for instance, let us mention the prediction of the evolution of epidemic diseases, or of the emergence of unsustainable economic inequalities), these questions are of special relevance in the framework of KTAP. In addition, the present paper is the first attempt to tackle them and to give some initial results about *both* the continuous dependence of solutions of Equation (1) on the coefficients *and* their instability. In connection with this last topic, the paper also offers some numerical simulations that show the separation between solutions corresponding to different values of parameters.

The contents of the paper are distributed as follows: in Section 2, we recall the structure of KTAP theory and Equation (1), and report the Cauchy problem associated with it, in the case in which the activity variable is assumed to be continuous (Section 2.1), as well as in the case in which it is assumed to be discrete (Section 2.2); Section 3 will be devoted to draw the notion of dependence of solutions on the parameters, and we state and prove a result concerning the continuous dependence of solutions on parameters, again in both the continuous case (Section 3.1) and the discrete case (Section 3.2), and, in Section 4, we give

first results about instability of solutions (in both cases); in this connection, special attention should be paid to the numerical simulations presented in Section 4.3 for the discrete case that offers a clear perception of the fact that the solutions to Equation (4) depend continuously on the parameters, but stay apart from each other when the perturbation of the parameters is greater than a well-defined threshold value. Finally, in Section 5, we outline some research perspectives based on some general and meaningful conclusions that can be drawn from the results found in the previous sections.

## 2. The Basic Equations

### 2.1. The Continuous Activity Framework

Let $D_u \subseteq \mathbb{R}$ and $F > 0$. According to what has been laid out in the Introduction, this paper is devoted to the analysis of properties of solutions $f(t, u) : [0, +\infty[ \times D_u \to \mathbb{R}^+$ of the following nonlinear integro-differential equation, with quadratic nonlinearity:

$$\partial_t f(t, u) + F \partial_u ((1 - u\, \mathbb{E}_1[f](t)) f(t, u)) = J[f, f](t, u), \tag{1}$$

where the operator $J[f, f](t, u)$ is defined as follows:

$$
\begin{aligned}
J[f, f](t, u) &= G[f, f](t, u) - L[f, f](t, u) \\
&= \int_{D_u \times D_u} \eta(u_*, u^*)\, \mathcal{A}(u_*, u^*, u)\, f(t, u_*) f(t, u^*)\, du_*\, du^* + \\
&\quad - f(t, u) \int_{D_u} \eta(u, u^*)\, f(t, u^*)\, du^*,
\end{aligned}
\tag{2}
$$

and

- $\eta(u_*, u^*) : D_u \times D_u \to \mathbb{R}^+$;
- $\mathcal{A}(u_*, u^*, u) : D_u \times D_u \times D_u \to \mathbb{R}^+$ with the property:

$$\int_{D_u} \mathcal{A}(u_*, u^*, u)\, du = 1, \qquad \forall u_*, u^* \in D_u;$$

- $\mathbb{E}_1[f](t) = \int_{D_u} u\, f(t, u)\, du.$

  The *Cauchy problem* associated with Equation (1) reads

$$
\begin{cases}
\partial_t f(t, u) + F \partial_u ((1 - u\, \mathbb{E}_1[f](t)) f(t, u)) = \\
J[f, f](t, u) & (t, u) \in [0, +\infty[ \times D_u \\
\\
f(0, u) = f^0(u) & u \in D_u.
\end{cases}
\tag{3}
$$

Let

$$\mathbb{E}_0[f](t) = \int_{D_u} f(t, u)\, du$$

and

$$\mathbb{E}_2[f](t) = \int_{D_u} u^2\, f(t, u)\, du.$$

Consider the *function space* $\mathcal{K}(D_u)$ defined as

$$\mathcal{K}(D_u) := \{ f(t, u) \in [0, +\infty[ \times D_u \to \mathbb{R}^+ : \mathbb{E}_0[f](t) = \mathbb{E}_2[f](t) = 1 \}.$$

The existence and uniqueness of solutions

$$f(t, u) \in C\left( (0, +\infty) \times D_u ; L^1(D_u) \right) \cap \mathcal{K}(D_u)$$

of the Cauchy problem (3) are proved in [25], under the condition

$$f(t, u) = 0, \qquad u \in \partial D_u.$$

The existence of solutions of the *nonequilibrium stationary problem* related to (1) is proved in [31]. A proof of the convergence of the solution of (3) to the nonequilibrium stationary solution as time goes to infinity is given in [32].

In many cases of interest, as for example in the description of the diffusion of epidemics $\eta(u_*, u^*)$ can be supposed to be constant, i.e., there exists $\eta > 0$ such that $\eta(u_*, u^*) = \eta$, for all $u_*, u^* \in D_u$.

**Remark 1.** *If $\mathcal{C}$ is a complex system, homogeneous with respect to the mechanical variables, i.e., space and velocity, (1) describes the evolution of the distribution function $f(t, u)$ of $\mathcal{C}$, and is called thermostatted kinetic framework [2].*

*The microscopic state is described by a scalar variable $u$, called activity, which attains its values in a real continuous subset $D_u$. In this frame:*

- *$\eta(u_*, u^*)$ is the interaction rate between the particles that are in the state $u_*$ and the particles in the state $u^*$;*
- *$\mathcal{A}(u_*, u^*, u)$ is the transition probability density i.e., the probability (density) that a particle in the state $u_*$ falls into the state $u$ after interacting with a particle in the state $u^*$;*
- *$F > 0$ is the value of the external force field acting on the system $\mathcal{C}$;*
- *$\mathbb{E}_0[f](t)$ is the density, $\mathbb{E}_1[f](t)$ is the linear momentum and $\mathbb{E}_2[f](t)$ is the global energy;*
- *$G[f, f](t, u)$ is the gain-term operator and $L[f, f](t, u)$ is the loss-term operator.*

*Equation (1) and the related problem (3) describe the evolution of a system $\mathcal{C}$ such that the global activation energy, $\mathbb{E}_2[f](t)$, is kept constant by means of a thermostat [33].*

### 2.2. The Discrete Activity Framework

Let $I_u = \{u_1, u_2, \dots, u_n\}$ be a discrete subset of $\mathbb{R}$. The operator $J_i[\mathbf{f}](t)$, for $i \in \{1, 2, \dots, n\}$ is defined as:

$$J_i[\mathbf{f}](t) = G_i[\mathbf{f}](t) - L_i[\mathbf{f}](t)$$
$$= \sum_{h=1}^{n} \sum_{k=1}^{n} \eta_{hk} B_{hk}^i f_h(t) f_k(t) - f_i(t) \sum_{k=1}^{n} \eta_{ik} f_k(t),$$

where $\eta_{hk} : I_u \times I_u \to \mathbb{R}^+$, for $h, k \in \{1, 2, \dots, n\}$, and the functions $B_{hk}^i : I_u \times I_u \times I_u \to \mathbb{R}^+$ (where $i, h, k \in \{1, 2, \dots, n\}$) obey the condition

$$\sum_{i=1}^{n} B_{hk}^i = 1, \quad h, k \in \{1, 2, \dots, n\}.$$

Let $\mathbf{f}(t) = (f_1(t), f_2(t), \dots, f_n(t))$, where, for any $i \in \{1, 2, \dots n\}$,

$$f_i(t) := f(t, u_i) : [0, +\infty[ \times I_u \to \mathbb{R}^+$$

is a solution of the *nonlinear ordinary differential equation*

$$\frac{df_i}{dt}(t) = J_i[\mathbf{f}](t) + F_i(t) - \sum_{i=1}^{n} \left( \frac{u_i^2 (J_i[\mathbf{f}] + F_i)}{\mathbb{E}_2[\mathbf{f}]} \right) f_i(t), \tag{4}$$

for $\mathbf{F}(t) = (F_1(t), F_2(t), \dots, F_n(t))$ with $F_i(t) > 0$. The 2-*nd order moment* function $\mathbb{E}_2[\mathbf{f}](t)$ of $\mathbf{f}$ takes now the form

$$\mathbb{E}_2[\mathbf{f}](t) = \sum_{i=1}^{n} u_i^2 f_i(t).$$

Consider the *function space*:

$$\mathcal{R}_{\mathbf{f}}^2 = \mathcal{R}_{\mathbf{f}}^2(\mathbb{R}^+; \mathbb{E}_2) = \left\{ \mathbf{f} \in C\left([0, +\infty]; \left(\mathbb{R}^+\right)^n\right) : \mathbb{E}_2[\mathbf{f}] = \mathbb{E}_2 \right\}$$

where $\mathbb{E}_2 \in \mathbb{R}^+$. The existence and uniqueness of solutions to the Cauchy problem associated with Equation (4), with initial data $\mathbf{f}^0$ such that $\sum_{i=1}^n u_i^2 f^0 = 1$, has been proved in [34] under the following assumption:

**H1** There exist $\eta, F > 0$, such that $F_i(t) \leq F$, for $t > 0$, and $\eta_{hk} \leq \eta$, for $h, k \in \{1, 2, \ldots, n\}$.

A *nonequilibrium stationary solution* of Equation (4), for $i \in \{1, 2, \ldots, n\}$, is a function $f_i$ satisfying the equation

$$J_i[\mathbf{f}] + F_i - \sum_{i=1}^n \left( \frac{u_i^2(J_i[\mathbf{f}] + F_i)}{\mathbb{E}_2} \right) f_i = 0. \tag{5}$$

Let $\tilde{\mathcal{R}}_{\mathbf{f}}^2$ denote the *function space*:

$$\tilde{\mathcal{R}}_{\mathbf{f}}^2(\mathbb{R}^+; \mathbb{E}_2) = \left\{ \mathbf{f} \in \left(\mathbb{R}^+\right)^n : \mathbb{E}_2[\mathbf{f}] = \mathbb{E}_2 \right\}.$$

The existence of *nonequilibrium stationary solutions* $g(u) \in \tilde{\mathcal{R}}_{\mathbf{f}}^2$ has been proved in [28], under the assumption **H1**.

In particular, under the further assumptions:

**H2** $\displaystyle\sum_{i=1}^n u_i B_{hk}^i = 0$, for all $h, k \in \{1, 2, \ldots, n\}$,

**H3** $\displaystyle\sum_{i=1}^n u_i^2 B_{hk}^i = u_h^2$, for all $h, k \in \{1, 2, \ldots, n\}$,

it has been proved in [28] that any nonequilibrium stationary solution is unique if the force field verifies the constraint

$$F > 2\eta \mathbb{E}_2^2 \left( 1 + \frac{1}{\|u\|_2^2} \right).$$

**Proposition 1** ([28]). *If assumptions **H1**–**H3** are met, together with the assumption*
**H4**

$$\mathbb{E}_0[\mathbf{f}] = \mathbb{E}_2[\mathbf{f}] = 1$$

*then*

1.  *The evolution equation of* $\mathbb{E}_1[\mathbf{f}](t) = \displaystyle\sum_{i=1}^n u_i f_i(t)$ *takes the form*

$$\mathbb{E}_1'[\mathbf{f}](t) + \left( \eta + \sum_{i=1}^n u_i^2 f_i \right) \mathbb{E}_1[\mathbf{f}](t) - \sum_{i=1}^n u_i F_i = 0; \tag{6}$$

2.  *as* $t \to +\infty$,

$$\mathbb{E}_1[\mathbf{f}](t) \to K := \frac{\displaystyle\sum_{i=1}^n u_i F_i}{\eta + \displaystyle\sum_{i=1}^n u_i^2 F_i}; \tag{7}$$

3.  *Denoting by* $\mathbf{f}_0$ *the initial data of the Cauchy problem related to* (4)*, one has*

$$|\mathbb{E}_1[\mathbf{f}](t) - K| \leq c \exp\left[ -\left( \eta + \sum_{i=1}^n u_i^2 F_i \right) t \right], \tag{8}$$

*where c is a constant depending on the system.*

**Remark 2.** *In [35], the existence of solutions of Equation (4) and of the related nonequilibrium stationary problem has been proved for more general values of the real discrete variable $u_i$.*

**Remark 3.** *Equation (4) has been proposed in [34] in order to model a complex system which partitioned in n subsystems, called* functional subsystems. *In particular:*

- *the function $f_i(t)$, for $i \in \{1, 2, \ldots, n\}$, denotes the* distribution function *of the i-th functional subsystem;*
- *the function $\mathbf{F}(t) = (F_1(t), F_2(t), \ldots, F_n(t))$ is the* external force field *acting on the whole system;*
- *The term*

$$\alpha := \sum_{i=1}^{n} \left( \frac{u_i^2 (J_i[\mathbf{f}] + F_i)}{\mathbb{E}_2} \right)$$

*represents the* thermostat term, *which allows for keeping constant the quantity $\mathbb{E}_2[\mathbf{f}](t)$:*
- *the term $\eta_{hk}$ is the* interaction rate *related to the encounters between the functional subsystem h and the functional subsystem k, for $h, k \in \{1, 2, \ldots, n\}$;*
- *the function $B_{hk}^i$ denotes the* transition probability density *that the functional subsystem h falls into the i after interacting with the functional subsystem k, for $i, h, k \in \{1, 2, \ldots, n\}$;*
- *the operator $J_i[\mathbf{f}](t)$, for $i \in \{1, 2, \ldots, n\}$, models the net flux to the i-th functional subsystem; $G_i[\mathbf{f}](t)$ denotes the* gain term operator *(incoming flux) and $L_i[\mathbf{f}](t)$ the* loss term operator *(outgoing flux).*

**Remark 4.** *Let $p \in \mathbb{N}$. Equation (4) can be further generalized as follows:*

$$\frac{df_i}{dt}(t) = J_i[\mathbf{f}](t) + F_i(t) - \sum_{i=1}^{n} \left( \frac{u_i^p (J_i[\mathbf{f}] + F_i)}{\mathbb{E}_p[\mathbf{f}]} \right) f_i(t).$$

*This framework allows for keeping the p-th order moment*

$$\mathbb{E}_p[\mathbf{f}](t) = \sum_{i=1}^{n} u_i^p f_i(t).$$

*of the distribution* **f** *constant.*

**Remark 5.** *The convergence of any solution of (4) to a corresponding nonequilibrium stationary state (solution to (5)), as time goes to infinity, has been proved in [27].*

## 3. The Continuous Dependence on the Parameters

### 3.1. The Continuous Activity Framework

Let $\mathcal{A}(u_*, u^*, u)$, $\tilde{\mathcal{A}}(u_*, u^*, u)$, $\eta$ and $\tilde{\eta}$ two classes of parameters for Equation (1). Let $J[f, f](t, u) = G[f, f](t, u) - L[f, f](t, u)$ be the operator related to the parameters $\mathcal{A}(u_*, u^*, u)$ and $\eta$, and $\tilde{J}[f, f](t, u) = \tilde{G}[f, f](t, u) - \tilde{L}[f, f](t, u)$ the operator related to the parameters $\tilde{\mathcal{A}}(u_*, u^*, u)$ and $\tilde{\eta}$.

The related Cauchy problems, with the same initial data $f^0(u)$, are defined as follows:

$$\begin{cases} \partial_t f(t, u) + F \partial_u ((1 - u \, \mathbb{E}_1[f](t)) f(t, u)) = J[f, f](t, u) \\ \\ f(0, u) = f^0(u), \end{cases} \tag{9}$$

$$\begin{cases} \partial_t f(t,u) + F\partial_u((1 - u\,\mathbb{E}_1[f](t))f(t,u)) = \tilde{J}[f,f](t,u) \\ \\ f(0,u) = f^0(u). \end{cases} \tag{10}$$

By [25], there exist two functions $f(t,u) \in C\big((0,+\infty) \times D_u; L^1(D_u)\big) \cap \mathcal{K}(D_u)$ and $\tilde{f}(t,u) \in C\big((0,+\infty) \times D_u; L^1(D_u)\big) \cap \mathcal{K}(D_u)$ that are solutions to problem (9) and to problem (10), respectively.

The present paper aims to give a contribution in two directions:

1. the *continuous dependence* of the solutions of Equation (1) on the *parameters* $\mathcal{A}(u_*, u^*, u)$ and $\eta$;
2. a first attempt towards the instability of the solutions of Equation (1) for certain values of the two classes of parameters $(\mathcal{A}(u_*, u^*, u), \eta)$ and $(\tilde{\mathcal{A}}(u_*, u^*, u), \tilde{\eta})$.

The first result will be a proof of the *continuous dependence* of solutions to Equation (1) with respect to the parameters.

Let $\Theta(u_*, u^*, u)$ be the function defined as:

$$\Theta(u_*, u^*, u) := |\eta\mathcal{A}(u_*, u^*, u) - \tilde{\eta}\tilde{A}(u_*, u^*, u)|.$$

**Theorem 1.** *Let* $f(t,u), \tilde{f}(t,u) \in C\big((0,+\infty) \times D_u; L^1(D_u)\big) \cap \mathcal{K}(D_u)$ *the solutions to problems* (9) *and* (10), *respectively. Assume that* $\Theta(u_*, u^*, u) \in L^1(D_u \times D_u \times D_u)$. *If there exist* $\delta, \hat{\delta} > 0$ *such that* $|\eta - \tilde{\eta}| < \delta$ *and* $\|\Theta(u_*, u^*, u)\|_{L^1(D_u \times D_u \times D_u)} \le \hat{\delta}$ *then, for all* $T > 0$:

$$\|f(t,u) - \tilde{f}(t,u)\|_{C\big((0,T) \times D_u; L^1(D_u)\big)} \le (\delta + \hat{\delta})\, T\, e^{(2\eta + \tilde{\eta})T}. \tag{11}$$

**Proof.** Integrating Equation (1) from 0 to $t$ and recalling that $f(0,u) = f^0(u)$ in mind, one has

$$\begin{aligned} f(t,u) = f^0(u) &+ \int_0^t J[f,f](\tau,u)\,d\tau \\ &- F\int_0^t \partial_u((1 - u\,\mathbb{E}_1[f](\tau))f(\tau,u))\,d\tau, \end{aligned} \tag{12}$$

where

$$\begin{aligned} \int_0^t J[f,f](\tau,u)\,d\tau &= \int_0^t G[f,f](\tau,u)\,d\tau - \int_0^t L[f,f](\tau,u)\,d\tau \\ &= \int_0^t \int_{D_u \times D_u} \eta\,\mathcal{A}(u_*, u^*, u)\,f(t,u_*)f(t,u^*)\,du_*\,du^*\,d\tau \\ &- \eta\int_0^t f(\tau,u)\,d\tau. \end{aligned}$$

The integral expression (12) for the solutions $f(t,u)$ and $\tilde{f}(t,u)$ becomes

$$\begin{aligned} f(t,u) = f^0(u) &+ \int_0^t \int_{D_u \times D_u} \eta\,\mathcal{A}(u_*, u^*, u)\,f(t,u_*)f(t,u^*)\,du_*\,du^*\,d\tau \\ &- \eta\int_0^t f(\tau,u)\,d\tau - F\int_0^t \partial_u((1 - u\,\mathbb{E}_1[f](\tau))f(\tau,u))\,d\tau, \end{aligned} \tag{13}$$

and

$$\begin{aligned} \tilde{f}(t,u) = f^0(u) &+ \int_0^t \int_{D_u \times D_u} \tilde{\eta}\,\tilde{\mathcal{A}}(u_*, u^*, u)\,\tilde{f}(t,u_*)\tilde{f}(t,u^*)\,du_*\,du^*\,d\tau \\ &- \tilde{\eta}\int_0^t \tilde{f}(\tau,u)\,d\tau - F\int_0^t \partial_u\big((1 - u\,\mathbb{E}_1[\tilde{f}](\tau))\tilde{f}(\tau,u)\big)\,d\tau \end{aligned} \tag{14}$$

respectively.

Subtracting side by side Equation (14) from Equation (13), we readily get

$$
\begin{aligned}
f(t,u) - \tilde{f}(t,u) &= \\
&= \int_0^t \int_{D_u \times D_u} \left( \eta \, \mathcal{A}(u_*, u^*, u) \, f(\tau, u_*) f(\tau, u^*) - \tilde{\eta} \, \tilde{\mathcal{A}}(u_*, u^*, u) \, \tilde{f}(\tau, u_*) \tilde{f}(\tau, u^*) \right) du_* \, du^* \, d\tau \\
&+ \int_0^t \left[ \tilde{\eta} \, \tilde{f}(\tau, u) - \eta \, f(\tau, u) \right] d\tau \\
&+ F \int_0^t \partial_u \left( \tilde{f}(\tau, u) - f(\tau, u) + f(\tau, u) \, u \, \mathbb{E}_1[f](\tau) - \tilde{f}(\tau, u) \, u \, \mathbb{E}_1[\tilde{f}](\tau) \right) d\tau
\end{aligned}
\tag{15}
$$

which leads immediately to the estimate

$$
\begin{aligned}
|f(t,u) - \tilde{f}(t,u)| &\leq \\
&\leq \left| \int_0^t \int_{D_u \times D_u} \eta \, \mathcal{A}(u_*, u^*, u) \, f(\tau, u_*) f(\tau, u^*) - \tilde{\eta} \, \tilde{\mathcal{A}}(u_*, u^*, u) \, \tilde{f}(\tau, u_*) \tilde{f}(\tau, u^*) \, du_* \, du^* \, d\tau \right| \\
&+ \left| \tilde{\eta} \int_0^t \tilde{f}(\tau, u) \, d\tau - \eta \int_0^t f(\tau, u) \, d\tau \right| \\
&+ F \left| \int_0^t \partial_u \left( \tilde{f}(\tau, u) - f(\tau, u) + f(\tau, u) \, u \, \mathbb{E}_1[f](\tau) - \tilde{f}(\tau, u) \, u \, \mathbb{E}_1[\tilde{f}](\tau) \right) d\tau \right|.
\end{aligned}
\tag{16}
$$

Since $f(t,u) = \tilde{f}(t,u) = 0$ for $u \in \partial D_u$, the third term on the right-hand side of inequality (16) vanishes, so that we obtain the relation

$$
\begin{aligned}
|f(t,u) - \tilde{f}(t,u)| &\leq \\
&\leq \left| \int_0^t \int_{D_u \times D_u} \eta \, \mathcal{A}(u_*, u^*, u) \, f(\tau, u_*) f(\tau, u^*) - \tilde{\eta} \, \tilde{\mathcal{A}}(u_*, u^*, u) \, \tilde{f}(\tau, u_*) \tilde{f}(\tau, u^*) \, du_* \, du^* \, d\tau \right| \\
&+ \int_0^t \left| \tilde{\eta} \, \tilde{f}(\tau, u) - \eta \, f(\tau, u) \right| d\tau
\end{aligned}
\tag{17}
$$

and, by straightforward calculations, one finds that the first term on the right-hand side of (17) can be estimated as follows:

$$
\begin{aligned}
\left| \int_0^t \int_{D_u \times D_u} \eta \, \mathcal{A}(u_*, u^*, u) \, f(\tau, u_*) f(\tau, u^*) - \tilde{\eta} \, \tilde{\mathcal{A}}(u_*, u^*, u) \, \tilde{f}(\tau, u_*) \tilde{f}(\tau, u^*) \, du_* \, du^* \, d\tau \right| &\leq \\
\leq \int_0^t \int_{D_u \times D_u} \eta \, f(\tau, u^*) \, \mathcal{A}(u_*, u^*, u) | f(\tau, u_*) - \tilde{f}(\tau, u_*)| \, du_* \, du^* \, d\tau \\
+ \int_0^t \int_{D_u \times D_u} \tilde{\eta} \, \tilde{f}(\tau, u_*) \, \tilde{\mathcal{A}}(u_*, u^*, u) | \tilde{f}(\tau, u^*) - f(\tau, u^*)| \, du^* \, du_* \, d\tau \\
+ \int_0^t \int_{D_u \times D_u} \tilde{f}(\tau, u_*) f(\tau, u^*) \left[ \eta \mathcal{A}(u_*, u^*, u) - \tilde{\eta} \tilde{\mathcal{A}}(u_*, u^*, u) \right] du_* \, du^* \, d\tau.
\end{aligned}
\tag{18}
$$

Now, using inequality (18) and integrating both sides of relation (17) on $D_u$, we get

$$
\begin{aligned}
\|f(t,u) - \tilde{f}(t,u)\|_{L^1(D_u)} &\leq \eta \int_0^t \|f(\tau,u) - \tilde{f}(\tau,u)\|_{L^1(D_u)} \, d\tau \\
&+ \tilde{\eta} \int_0^t \|f(\tau,u) - \tilde{f}(\tau,u)\|_{L^1(D_u)} \, d\tau + \|\Theta(u_*, u^*, u)\|_{L^1(D_u \times D_u \times D_u)} \\
&+ \int_0^t \int_{D_u} |\tilde{\eta} \, \tilde{f}(\tau, u) - \eta \, f(\tau, u)| \, du \, d\tau.
\end{aligned}
\tag{19}
$$

Since:

$$
\begin{aligned}
|\tilde{\eta} \, \tilde{f}(\tau, u) - \eta \, f(\tau, u)| &= |\tilde{\eta} \, \tilde{f}(\tau, u) - \eta \, \tilde{f}(\tau, u) + \eta \, \tilde{f}(\tau, u) - \eta \, f(\tau, u)| \\
&\leq \tilde{f}(\tau, u) |\tilde{\eta} - \eta| + \eta |\tilde{f}(\tau, u) - f(\tau, u)|,
\end{aligned}
$$

relation (19), bearing in mind that $|\eta - \tilde{\eta}| < \delta$ and $\|\Theta(u_*, u^*, u)\|_{L^1(D_u \times D_u \times D_u)} \leq \hat{\delta}$, may be rewritten in the form

$$
\begin{aligned}
\|f(t,u) - \tilde{f}(t,u)\|_{L^1(D_u)} \leq & \int_0^t (\eta + \tilde{\eta}) \|f(\tau, u) - \tilde{f}(\tau, u)\|_{L^1(D_u)} \, d\tau \\
& + \|\Theta(u_*, u^*, u)\|_{L^1(D_u \times D_u \times D_u)} + |\tilde{\eta} - \eta| t \\
& + \eta \int_0^t \|f(\tau, u) - \tilde{f}(\tau, u)\|_{L^1(D_u)} \, d\tau \\
\leq & \int_0^t (2\eta + \tilde{\eta}) \|f(\tau, u) - \tilde{f}(\tau, u)\|_{L^1(D_U)} \, d\tau + (\delta + \hat{\delta}) t.
\end{aligned}
\tag{20}
$$

By Grönwall's inequaility [36],

$$
\|f(t,u) - \tilde{f}(t,u)\|_{L^1(D_u)} \leq (\delta + \hat{\delta}) t \, e^{(2\eta + \tilde{\eta})t}. \tag{21}
$$

By (20) and (21), relation (11) is proved, i.e., for $T > 0$,

$$
\|f(t,u) - \tilde{f}(t,u)\|_{C\left((0,T) \times D_u; L^1(D_u)\right)} \leq (\delta + \hat{\delta}) T \, e^{(2\eta + \tilde{\eta})T}.
$$

□

**Remark 6.** *It is worth pointing out that*

1. *the assumption $|\eta - \tilde{\eta}| < \delta$ is an estimate of the distance between the interaction rates;*
2. *the assumption $\|\Theta(u_*, u^*, u)\|_{L^1(D_u \times D_u \times D_u)} \leq \hat{\delta}$ is an estimate on the distance between the transition probability densities, "weighted" by the interaction rates.*

**Remark 7.** *The conclusion (11) of Theorem 1 ensures the continuous dependence of solutions on the parameters $\mathcal{A}(u_*, u^*, u)$ and $\eta$. Indeed,*

$$
\|f(t,u) - \tilde{f}(t,u)\|_{C\left((0,+\infty) \times D_u; L^1(D_u)\right)} \xrightarrow{\delta, \hat{\delta} \to 0} 0.
$$

*3.2. The Discrete Activity Framework*

This section aims to prove the continuous dependence of the solutions of Equation (4) on the parameters $\eta_{hk}$ and $B_{hk}^i$ when $\mathbf{f}^0$, $\mathbf{F}(t) = \mathbf{F}$, constant in time, and $T > 0$ are fixed.

Let $\mathbf{f}(t) = (f_1(t), f_2(t), \ldots, f_n(t)), \hat{\mathbf{f}}(t) = \left(\hat{f}_1(t), \hat{f}_2(t), \ldots, \hat{f}_n(t)\right)$ be the solutions of the systems

$$
\begin{cases}
\dfrac{df_i}{dt}(t) = J_i[\mathbf{f}](t) + F_i(t) - \displaystyle\sum_{i=1}^n \left( \dfrac{u_i^2 (J_i[\mathbf{f}] + F_i)}{\mathbb{E}_2[\mathbf{f}]} \right) f_i(t) & t \in [0, T] \\[4mm]
\mathbf{f}(0) = \mathbf{f}^0,
\end{cases}
\tag{22}
$$

and

$$
\begin{cases}
\dfrac{df_i}{dt}(t) = \hat{J}_i[\mathbf{f}](t) + F_i(t) - \displaystyle\sum_{i=1}^n \left( \dfrac{u_i^2 (\hat{J}_i[\mathbf{f}] + F_i)}{\mathbb{E}_2[\mathbf{f}]} \right) f_i(t) & t \in [0, T] \\[4mm]
\mathbf{f}(0) = \mathbf{f}^0,
\end{cases}
\tag{23}
$$

respectively; the operators $\mathbf{J}[\mathbf{f}]$ and $\hat{\mathbf{J}}[\mathbf{f}]$ are defined by the parameters $\eta_{hk}$, $B_{hk}^i$, and $\hat{\eta}_{hk}$, $\hat{B}_{hk}^i$, respectively.

The following stability result holds.

**Theorem 2.** *Let* $\mathbf{f}(t) = (f_1(t), f_2(t), \ldots, f_n(t)), \hat{\mathbf{f}}(t) = (\hat{f}_1(t), \hat{f}_2(t), \ldots, \hat{f}_n(t))$ *be the solutions of* (12) *and* (13), *respectively. Assume* $\eta_{hk} \leq \eta$, $\hat{\eta}_{hk} \leq \hat{\eta}$, *for* $h, k \in \{1, 2, \ldots, n\}$, *and* $F_i \leq F$, *for* $i \in \{1, 2, \ldots, n\}$, *for* $\eta, \hat{\eta}, F > 0$. *If* $\Lambda := \sum_{i=1}^{n} \sum_{h,k=1}^{n} \left| \eta_{hk} B_{hk}^i - \hat{\eta}_{hk} \hat{B}_{hk}^i \right|$, *then*

$$\max_{t \in [0,T]} \left\| \mathbf{f}(t) - \hat{\mathbf{f}}(t) \right\|_1 \leq \Lambda \, T \, e^{\left( \eta + \hat{\eta} + \sum_{i=1}^{n} u_i^2 F_i \right) T} \tag{24}$$

*where*

$$\| \mathbf{f}(t) - \hat{\mathbf{f}}(t) \|_1 := \sum_{i=1}^{n} |f_i(t) - \hat{f}_i(t)|.$$

**Proof.** Bearing assumption **H4** in mind, and integrating Equations (22) and (23) on $[0, t]$, we get

$$f_i(t) = f_i^0 + \int_0^t \left( J_i[\mathbf{f}](t) + F_i - \left( \sum_{i=1}^{n} u_i^2 (J_i[\mathbf{f}](t) + F_i) \right) f_i(t) \right) dt, \tag{25}$$

and

$$\hat{f}_i(t) = f_i^0 + \int_0^t \left( \hat{J}_i[\mathbf{f}](t) + F_i - \left( \sum_{i=1}^{n} u_i^2 \left( \hat{J}_i[\hat{\mathbf{f}}](t) + F_i \right) \right) \hat{f}_i(t) \right) dt \tag{26}$$

for $i \in \{1, 2, \ldots, n\}$. Now, subtracting (26) from (25), we find

$$f_i(t) - \hat{f}_i(t) =$$
$$= \int_0^t J_i[\mathbf{f}](t) - \hat{J}_i[\hat{\mathbf{f}}](t) \, dt \tag{27}$$
$$- \int_0^t \left[ \left( \sum_{i=1}^{n} u_i^2 (J_i[\mathbf{f}](t) + F_i) \right) f_i(t) - \left( \sum_{i=1}^{n} u_i^2 \left( \hat{J}_i[\hat{\mathbf{f}}](t) + F_i \right) \hat{f}_i(t) \right) \right] dt.$$

By taking the side-by-side sum on $i \in \{1, 2, \ldots, n\}$ of these last relations, we arrive at

$$\sum_{i=1}^{n} \left| f_i(t) - \hat{f}_i(t) \right| \leq$$

$$\leq \int_0^t \sum_{i=1}^{n} \left| J_i[\mathbf{f}](t) - \hat{J}_i[\hat{\mathbf{f}}](t) \right| \tag{28}$$
$$+ \int_0^t \sum_{i=1}^{n} \left| \left( \sum_{i=1}^{n} u_i^2 (J_i[\mathbf{f}](t) + F_i) \right) f_i(t) - \left( \sum_{i=1}^{n} u_i^2 \left( \hat{J}_i[\hat{\mathbf{f}}](t) + F_i \right) \hat{f}_i(t) \right) \right| dt.$$

Now, first of all, observe that

$$\int_0^t \sum_{i=1}^{n} |J_i[\mathbf{f}](t) - \hat{J}_i[\hat{\mathbf{f}}](t)| \leq$$

$$\leq \int_0^t \sum_{i=1}^{n} \left| \sum_{h,k=1}^{n} \left( \eta_{hk} B_{hk}^i f_h(t) f_k(t) - \hat{\eta}_{hk} \hat{B}_{hk}^i \hat{f}_h(t) \hat{f}_k(t) \right) \right| dt \tag{29}$$
$$+ \int_0^t \sum_{i=1}^{n} \left| f_i(t) \sum_{k=1}^{n} \eta_{ik} f_k(t) - \hat{f}_i(t) \sum_{k=1}^{n} \hat{\eta}_{ik} \hat{f}_k(t) \right|.$$

The integrand in the first term on the right-hand side of (29) can be estimated as follows:

$$
\begin{aligned}
&\left| \eta_{hk} B_{hk}^i f_h(t) f_k(t) - \hat{\eta}_{hk} \hat{B}_{hk}^i \hat{f}_h(t) \hat{f}_k(t) \right| = \\
&= \left| \eta_{hk} B_{hk}^i f_h(t) f_k(t) - B_{hk}^i \eta_{hk} f_h(t) \hat{f}_k(t) + B_{hk}^i \eta_{hk} f_h(t) \hat{f}_k(t) - \hat{\eta}_{hk} \hat{B}_{hk}^i \hat{f}_h(t) \hat{f}_k(t) \right| \\
&\leq \left| \eta_{hk} B_{hk}^i f_h(t) \left( f_k(t) - \hat{f}_k(t) \right) + \hat{f}_k(t) \left( \eta_{hk} B_{hk}^i f_h(t) - \hat{\eta}_{hk} \hat{B}_{hk}^i \hat{f}_h(t) \right) \right| \\
&\leq \eta_{hk} B_{hk}^i f_h(t) \left| f_k(t) - \hat{f}_k(t) \right| \\
&\quad + \hat{f}_k(t) \left| \eta_{hk} B_{hk}^i f_h(t) - \hat{\eta}_{hk} \hat{B}_{hk}^i f_h(t) + \hat{\eta}_{hk} \hat{B}_{hk}^i f_h(t) - \hat{\eta}_{hk} \hat{B}_{hk}^i \hat{f}_h(t) \right| \\
&\leq \eta_{hk} B_{hk}^i f_h(t) \left| f_k(t) - \hat{f}_k(t) \right| + \hat{f}_k(t) f_h(t) \left| \eta_{hk} B_{hk}^i - \hat{\eta}_{hk} \hat{B}_{hk}^i \right| \\
&\quad + \hat{\eta}_{hk} \hat{B}_{hk}^i \left| f_h(t) - \hat{f}_h(t) \right|.
\end{aligned}
\tag{30}
$$

By using this estimate, the first integral on the right-hand side of (29) turns out to be majorized as follows:

$$
\begin{aligned}
&\int_0^t \sum_{i=1}^n \left| \sum_{h,k=1}^n \left( \eta_{hk} B_{hk}^i f_h(t) f_k(t) - \hat{\eta}_{hk} \hat{B}_{hk}^i \hat{f}_h(t) \hat{f}_k(t) \right) \right| dt \\
&\leq \int_0^t \sum_{i=1}^n \sum_{h,k=1}^n \eta_{hk} B_{hk}^i f_h(t) \left| f_k(t) - \hat{f}_k(t) \right| dt + \int_0^t \sum_{i=1}^n \sum_{h,k=1}^n \hat{\eta}_{hk} \hat{B}_{hk}^i \hat{f}_k(t) \left| f_h(t) - \hat{f}_h(t) \right| dt \\
&\quad + \int_0^t \sum_{h,k=1}^n \hat{f}_k(t) f_h(t) \sum_{i=1}^n \left| \eta_{hk} B_{hk}^i - \hat{\eta}_{hk} \hat{B}_{hk}^i \right| dt \\
&\leq \eta \int_0^t \| \mathbf{f}(t) - \hat{\mathbf{f}}(t) \|_1 \, dt + \hat{\eta} \int_0^t \| \mathbf{f}(t) - \hat{\mathbf{f}}(t) \|_1 \, dt + \Lambda \, t.
\end{aligned}
\tag{31}
$$

As far as the second term on the right-hand side of (28) is concerned, one has

$$
\begin{aligned}
\sum_{i=1}^n u_i^2 J_i[\mathbf{f}](t) &= \sum_{i=1}^n u_i^2 (G_i[\mathbf{f}](t) - L_i[\mathbf{f}](t)) \\
&= \sum_{i=1}^n u_i^2 \left( \sum_{h,k=1}^n \eta_{hk} B_{hk}^i f_h(t) f_k(t) - f_i(t) \sum_{k=1}^n \eta_{ik} f_k(t) \right) \\
&= \sum_{h,k=1}^n \left( \sum_{i=1}^n u_i^2 B_{hk}^i \right) \eta_{hk} f_h(t) f_k(t) - \sum_{i=1}^n u_i^2 f_i(t) \sum_{k=1}^n \eta_{ik} f_k(t) = 0
\end{aligned}
\tag{32}
$$

which in turn implies

$$
\begin{aligned}
&\int_0^t \sum_{i=1}^n \left| \left( \sum_{i=1}^n u_i^2 (J_i[\mathbf{f}](t) + F_i) \right) f_i(t) - \left( \sum_{i=1}^n u_i^2 \left( \hat{J}_i[\hat{\mathbf{f}}](t) + F_i \right) \hat{f}_i(t) \right) \right| dt \leq \\
&\leq \int_0^t \left( \sum_{i=1}^n u_i^2 F_i \right) \| \mathbf{f}(t) - \hat{\mathbf{f}}(t) \|_1 \, dt.
\end{aligned}
\tag{33}
$$

Finally, by using relations (31) and (33), inequality (28) becomes

$$
\| \mathbf{f}(t) - \hat{\mathbf{f}}(t) \|_1 \leq \int_0^t \left( \eta \, h + \hat{\eta} \, k + \sum_{i=1}^n u_i^2 F_i \right) \| \mathbf{f}(t) - \hat{\mathbf{f}}(t) \|_1 \, dt + \Lambda \, t
\tag{34}
$$

and now Grönwall's inequaility [36] yields

$$
\| \mathbf{f}(t) - \hat{\mathbf{f}}(t) \|_1 \leq \Lambda \, T \, e^{\left( \eta \, h + \hat{\eta} \, k + \sum_{i=1}^n u_i^2 F_i \right) t}
$$

leading at once to (24).    □

**Remark 8.** *The conclusion of Theorem 2 is the continuous dependence of solution of Equation (4) on the parameters of the system, i.e., the interaction rate $\eta_{hk}$ and the transition probability density $B^i_{hk}$. In fact,*

$$\max_{t \in [0,T]} \|\mathbf{f}(t) - \hat{\mathbf{f}}(t)\|_1 \xrightarrow{\Lambda \to 0} 0.$$

**Remark 9.** *The coefficient $\Lambda$ defined in Theorem 2 is a first estimate of the distance between the two classes of parameters, i.e., $(\eta_{hk}, B^i_{hk})$ and $(\hat{\eta}_{hk}, \hat{B}^i_{hk})$.*

## 4. A First Attempt towards the Instability with Respect to the Parameters

### 4.1. The Continuous Activity Framework

This section aims to give a first result about instability of solutions of Equation (1) with respect to the parameters, interaction rate $\eta$, and transition probability density $\mathcal{A}(u_*, u^*, u)$.

**Theorem 3.** *Let $f(t, u), \tilde{f}(t, u) \in C\big((0, +\infty) \times D_u; L^1(D_u)\big) \cap \mathcal{K}(D_u)$ be the solutions to problems (9) and (10), respectively. Assume that $\Theta(u_*, u^*, u) \in L^1(D_u \times D_u \times D_u)$. If there exist two constants $M_1$ and $\hat{M}_1$, with $M_1 > \hat{M}_1$ such that $|\eta - \tilde{\eta}| > M_1$ and $\|\Theta(u_*, u^*, u)\|_{L^1(D_u \times D_u \times D_u)} \leq \hat{M}_1$, then, for all $T > 0$:*

$$\|f(t,u) - \tilde{f}(t,u)\|_{C\big((0,T) \times D_u; L^1(D_u)\big)} \geq \frac{(M_1 - \hat{M}_1)}{1 + (\eta + \tilde{\eta})T} T > 0. \tag{35}$$

**Proof.** As in Theorem 1, by using the integral formulation of (1) and by straightforward calculations, one has

$$
\begin{aligned}
|f(t,u) - \tilde{f}(t,u)| = \bigg| &\int_0^t \big(\tilde{\eta}\, \tilde{f}(\tau, u) - \eta\, f(\tau, u)\big)\, d\tau \\
&+ \int_0^t \int_{D_u \times D_u} \eta\, \mathcal{A}(u_*, u^*, u)\, f(\tau, u_*) f(\tau, u^*) - \tilde{\eta}\, \tilde{A}(u_*, u^*, u)\, \tilde{f}(\tau, u_*) \tilde{f}(\tau, u^*)\, du_*\, du^*\, d\tau \\
&+ F \int_0^t \partial_u \big((\tilde{f}(\tau, u) - f(\tau, u) + f(\tau, u)\, u\, \mathbb{E}_1[f](\tau) - \tilde{f}(\tau, u)\, u\, \mathbb{E}_1[\tilde{f}](\tau))\big)\, d\tau \bigg| \\
\geq &\bigg| \int_0^t \big(\tilde{\eta}\, \tilde{f}(\tau, u) - \eta\, f(\tau, u)\big)\, d\tau \bigg| \\
&- \bigg| \int_0^t \int_{D_u \times D_u} \eta\, \mathcal{A}(u_*, u^*, u)\, f(\tau, u_*) f(\tau, u^*) - \tilde{\eta}\, \tilde{A}(u_*, u^*, u)\, \tilde{f}(\tau, u_*) \tilde{f}(\tau, u^*)\, du_*\, du^*\, d\tau \bigg| \\
&- F \bigg| \int_0^t \partial_u \big((\tilde{f}(\tau, u) - f(\tau, u) + f(\tau, u)\, u\, \mathbb{E}_1[f](\tau) - \tilde{f}(\tau, u)\, u\, \mathbb{E}_1[\tilde{f}](\tau))\big)\, d\tau \bigg|
\end{aligned}
\tag{36}
$$

whence, by integrating the (36) on $D_u$, we obtain

$$
\begin{aligned}
\|f(t,u) - \tilde{f}(t,u)\|_{L^1(D_u)} \geq &\int_{D_u} \bigg| \int_0^t \big(\tilde{\eta}\, \tilde{f}(\tau, u) - \eta\, f(\tau, u)\big)\, d\tau \bigg|\, du \\
&- \int_{D_u} \bigg| \int_0^t \int_{D_u \times D_u} \eta\, \mathcal{A}(u_*, u^*, u)\, f(\tau, u_*) f(\tau, u^*) - \tilde{\eta}\, \tilde{A}(u_*, u^*, u)\, \tilde{f}(\tau, u_*) \tilde{f}(\tau, u^*)\, du_*\, du^*\, d\tau \bigg|\, du \\
&- F \int_{D_u} \bigg| \int_0^t \partial_u \big((\tilde{f}(\tau, u) - f(\tau, u) + f(\tau, u)\, u\, \mathbb{E}_1[f](\tau) - \tilde{f}(\tau, u)\, u\, \mathbb{E}_1[\tilde{f}](\tau))\big)\, d\tau \bigg|.
\end{aligned}
\tag{37}
$$

Since $f(t, u) = \tilde{f}(t, u) = 0$ for $u \in \partial D_u$, the third term on the right-hand side of the (37) vanishes.

The first term at the right-hand side of inequality (37) is estimated as follows:

$$\int_{D_u} \left| \int_0^t \left( \tilde{\eta} \, \tilde{f}(\tau, u) - \eta \, f(\tau, u) \right) d\tau \right| du \geq \left| \int_0^t \int_{D_u} \tilde{\eta} \, \tilde{f}(\tau, u) - \eta \, f(\tau, u) \, du \, d\tau \right|$$

$$= |\tilde{\eta} - \eta| \, t. \tag{38}$$

Consider now the second term of the right-hand side of inequality (37). First of all,

$$-\int_{D_u} \left| \int_0^t \int_{D_u \times D_u} \eta \, \mathcal{A}(u_*, u^*, u) \, f(\tau, u_*) f(\tau, u^*) \right.$$

$$\left. - \tilde{\eta} \, \tilde{A}(u_*, u^*, u) \, \tilde{f}(\tau, u_*) \tilde{f}(\tau, u^*) \, du_* \, du^* \, d\tau \right| du \geq$$

$$-\int_{D_u} \int_0^t \int_{D_u \times D_u} \left| \eta \, \mathcal{A}(u_*, u^*, u) \, f(\tau, u_*) f(\tau, u^*) \right.$$

$$\left. - \tilde{\eta} \, \tilde{A}(u_*, u^*, u) \, \tilde{f}(\tau, u_*) \tilde{f}(\tau, u^*) \right| du_* \, du^* \, d\tau \, du. \tag{39}$$

By straightforward calculations,

$$\left| \eta \, \mathcal{A}(u_*, u^*, u) \, f(\tau, u_*) f(\tau, u^*) - \tilde{\eta} \, \tilde{A}(u_*, u^*, u) \, \tilde{f}(\tau, u_*) \tilde{f}(\tau, u^*) \right| =$$

$$= \left| \eta \, \mathcal{A}(u_*, u^*, u) \, f(\tau, u_*) \, f(\tau, u^*) - \eta \, \mathcal{A}(u_*, u^*, u) \, f(\tau, u_*) \tilde{f}(\tau, u^*) \right.$$

$$+ \eta \, \mathcal{A}(u_*, u^*, u) \, f(\tau, u_*) \tilde{f}(\tau, u^*) + \tilde{\eta} \, \tilde{A}(u_*, u^*, u) \, f(\tau, u_*) \tilde{f}(\tau, u^*)$$

$$- \tilde{\eta} \, \tilde{A}(u_*, u^*, u) \, f(\tau, u_*) \tilde{f}(\tau, u^*) - \tilde{\eta} \, \tilde{A}(u_*, u^*, u) \, \tilde{f}(\tau, u_*) \tilde{f}(\tau, u^*) \right|$$

$$= \left| \eta \, \mathcal{A}(u_*, u^*, u) \, f(\tau, u_*) \big( f(\tau, u^*) - \tilde{f}(\tau, u^*) \big) \right.$$

$$- \tilde{\eta} \, \tilde{A}(u_*, u^*, u) \, \tilde{f}(\tau, u^*) \big( f(\tau, u_* - f(\tau, u_*)) \big)$$

$$+ f(\tau, u_*) \tilde{f}(\tau, u^*) \big( \eta \, \mathcal{A}(u_*, u^*, u) - \tilde{\eta} \, \tilde{A}(u_*, u^*, u) \big) \right|$$

$$\leq \eta \, \mathcal{A}(u_*, u^*, u) \, f(\tau, u_*) \big| f(\tau, u^*) - \tilde{f}(\tau, u_*) \big|$$

$$+ \tilde{\eta} \, \tilde{A}(u_*, u^*, u) \, \tilde{f}(\tau, u^*) \big| f(\tau, u_*) - \tilde{f}(\tau, u_*) \big|$$

$$+ f(\tau, u_*) \, \tilde{f}(\tau, u^*) \big| \eta \, \mathcal{A}(u_*, u^*, u) - \tilde{\eta} \tilde{A}(u_*, u^*, u) \big|. \tag{40}$$

In virtue of inequalities (39) and (40),

$$-\int_{D_u} \left| \int_0^t \int_{D_u \times D_u} \eta \, \mathcal{A}(u_*, u^*, u) \, f(\tau, u_*) f(\tau, u^*) \right.$$

$$\left. - \tilde{\eta} \, \tilde{A}(u_*, u^*, u) \, \tilde{f}(\tau, u_*) \tilde{f}(\tau, u^*) \, du_* \, du^* \, d\tau \right| du \geq$$

$$\geq -\eta \int_0^t \int_{D_u} f(\tau, u_*) \int_{D_u} |f(\tau, u^*) - \tilde{f}(\tau, u^*)| \int_{D_u} \mathcal{A}(u_*, u^*, u) \, du \, du^* \, du_* \, d\tau$$

$$- \tilde{\eta} \int_0^t \int_{D_u} \tilde{f}(\tau, u^*) \int_{D_u} |f(\tau, u_*) - \tilde{f}(\tau, u_*)| \int_{D_u} \mathcal{A}(u_*, u^*, u) \, du \, du_* \, du^* \, d\tau$$

$$- \int_0^t \int_{D_u} \int_{D_u \times D_u} f(\tau, u_*) \tilde{f}(\tau, u^*) \big| \eta \, \mathcal{A}(u_*, u^*, u) - \tilde{\eta} \, \tilde{A}(u_*, u^*, u) \big| \, du_* \, du^* \, du \, d\tau$$

$$= -\eta \, t \| f(t, u) - \tilde{f}(t, u) \|_{L^1(D_u)} - \tilde{\eta} \, t \| f(t, u) - \tilde{f}(t, u) \|_{L^1(D_u)}$$

$$- \int_0^t \int_{D_u} \int_{D_u \times D_u} f(\tau, u_*) \tilde{f}(\tau, u^*) \big| \eta \, \mathcal{A}(u_*, u^*, u) - \tilde{\eta} \, \tilde{A}(u_*, u^*, u) \big| \, du_* \, du^* \, du \, d\tau \tag{41}$$

and, using Hölder's inequality,

$$\int_0^t \int_{D_u} \int_{D_u \times D_u} \tilde{f}(\tau, u^*) f(\tau, u_*) \big| \eta\, \mathcal{A}(u_*, u^*, u) - \tilde{\eta}\, \tilde{A}(u_*, u^*, u) \big|\, du_*\, du^*\, du\, d\tau \leq$$

$$\leq \int_0^t \int_{D_u} \int_{D_u} \tilde{f}(\tau, u^*) \left( \max_{u_* \in D_u} f(\tau, u_*) \right) \int_{D_u} \big| \eta\, \mathcal{A}(u_*, u^*, u) - \tilde{\eta}\, \tilde{A}(u_*, u^*, u) \big|\, du_*\, du^*\, du\, d\tau$$

$$\leq \int_0^t \int_{D_u} \int_{D_u} \left( \max_{u^* \in D_u} f(\tau, u^*) \right) \int_{D_u} \big| \eta\, \mathcal{A}(u_*, u^*, u) - \tilde{\eta}\, \tilde{A}(u_*, u^*, u) \big|\, du^*\, du_*\, du\, d\tau \tag{42}$$

$$\leq \int_0^t \int_{D_u \times D_u \times D_u} \big| \eta\, \mathcal{A}(u_*, u^*, u) - \tilde{\eta}\, \tilde{A}(u_*, u^*, u) \big|\, du_*\, du^*\, du$$

$$= \| \Theta(u_*, u^*, u) \|_{L^1(D_u \times D_u \times D_u)} t.$$

Thanks to relations (41) and (42), inequality (39) becomes

$$- \int_{D_u} \left| \int_0^t \int_{D_u \times D_u} \eta\, \mathcal{A}(u_*, u^*, u)\, f(\tau, u_*) f(\tau, u^*) \right.$$
$$\left. - \tilde{\eta}\, \tilde{A}(u_*, u^*, u)\, \tilde{f}(\tau, u_*) \tilde{f}(\tau, u^*)\, du_*\, du^*\, d\tau \right| du \geq \tag{43}$$
$$\geq -(\eta + \tilde{\eta}) t \| f(t, u) - \tilde{f}(t, u) \|_{L^1(D_u)} - \| \Theta(u_*, u^*, u) \|_{L^1(D_u \times D_u \times D_u)} t.$$

Finally, by (38) and (43), inequality (37) yields

$$(1 + (\eta + \tilde{\eta}) t) \| f(t, u) - \tilde{f}(t, u) \|_{L^1(D_u)} \geq |\tilde{\eta} - \eta|\, t - \| \Theta(u_*, u^*, u) \|_{L^1(D_u \times D_u \times D_u)} t.$$

Then:

$$\| f(t, u) - \tilde{f}(t, u) \|_{L^1(D_u)} \geq \frac{|\tilde{\eta} - \eta| - \| \Theta(u_*, u^*, u) \|_{L^1(D_u \times D_u \times D_u)}}{1 + (\eta + \tilde{\eta}) t}\, t. \tag{44}$$

Relation (35) is then proved by using the (44), and keeping in mind the fact that $|\eta - \tilde{\eta}| > M_1$ and $\| \Theta(u_*, u^*, u) \|_{L^1(D_u \times D_u \times D_u)} \leq \hat{M}_1$, with $M_1 > \hat{M}_1$:

$$\| f(t, u) - \tilde{f}(t, u) \|_{C\big((0,T) \times D_u; L^1(D_u)\big)} \geq \frac{|\tilde{\eta} - \eta| - \| \Theta(u_*, u^*, u) \|_{L^1(D_u \times D_u \times D_u)}}{1 + (\eta + \tilde{\eta}) T}\, T$$

$$\geq \frac{(M_1 - \hat{M}_1)}{1 + (\eta + \tilde{\eta}) T}\, T.$$

□

**Remark 10.** *In Theorem 3, the instability is related to the variation of the interaction rate.*

**Remark 11.** *For instance, if $D_u = [0, \frac{1}{2}]$ is taken into account with $\mathcal{A} = \tilde{\mathcal{A}}$, then the right-hand side of relation (35) is strictly positive, so that the instability of the solutions follows at once.*

*4.2. The Discrete Activity Framework*

In this section, we want to outline a first step of a study of instability in the discrete framework (4). This is an important issue in view of future numerical analysis.

**Theorem 4.** *Let $\mathbf{f}(t) = (f_1(t), f_2(t), \ldots, f_n(t))$, $\hat{\mathbf{f}}(t) = (\hat{f}_1(t), \hat{f}_2(t), \ldots, \hat{f}_n(t))$ be the solutions of Equations (22) and (23), respectively. Let $\eta, \hat{\eta}, F \geq 0$ such that $\eta_{hk} \leq \eta$, $\hat{\eta}_{hk} \leq \hat{\eta}$, for $h, k \in \{1, 2, \ldots, n\}$, and $F_i \leq F$, for $i \in \{1, 2, \ldots, n\}$. Furthermore, let*

$$\Gamma(t) := \min_{\mathbf{f}, \hat{\mathbf{f}} \in (C([0,T]))^n} \left\{ \sum_{i=1}^n \left| \int_0^t \sum_{h,k=1}^n \eta_{hk}\, B_{hk}^i\, f_h(t)\, f_k(t) - \hat{\eta}_{hk}\, \hat{B}_{hk}^i\, \hat{f}_h(t)\, \hat{f}_k(t) \right| \right\}.$$

*Then,*

$$\max_{[0,T]} \|\mathbf{f}(t) - \hat{\mathbf{f}}(t)\|_1 \geq \max_{[0,T]} \frac{\Gamma(t) - (\eta + \hat{\eta})t}{1 + (\sum_{i=1}^{n} u_i^2 F_i)t}. \tag{45}$$

**Proof.** Bearing the (27) in mind, and by using the (32), straightforward calculations show, for $i \in \{1, 2, \ldots, n\}$:

$$
\begin{aligned}
\left| f_i(t) - \hat{f}_i(t) \right| &\geq \\
&\geq \left| \int_0^t \sum_{h,k=1}^{n} \eta_{hk} B_{hk}^i f_h(s) f_k(s) - \hat{\eta}_{hk} \hat{B}_{hk}^i \hat{f}_h(s) \hat{f}_k(s) \, ds \right| \\
&\quad - \left| \int_0^t \left( \hat{f}_i(s) \sum_{k=1}^{n} \hat{\eta}_{ik} \hat{f}_k(s) - f_i(s) \sum_{k=1}^{n} \eta_{ik} f_k(s) \right) ds \right| \\
&\quad - \left( \sum_{i=1}^{n} u_i^2 F_i \right) \left| \int_0^t \left( \hat{f}_i(s) - f_i(s) \right) ds \right|.
\end{aligned}
\tag{46}
$$

By taking the sum on $i \in 1, 2, \ldots, n$ of relations (46), we find

$$
\begin{aligned}
\|\mathbf{f}(t) - \hat{\mathbf{f}}(t)\|_1 &\geq \\
&\geq \sum_{i=1}^{n} \left| \int_0^t \sum_{h,k=1}^{n} \eta_{hk} B_{hk}^i f_h(s) f_k(s) - \hat{\eta}_{hk} \hat{B}_{hk}^i \hat{f}_h(s) \hat{f}_k(s) \, ds \right| \\
&\quad - \sum_{i=1}^{n} \left| \int_0^t \left( \hat{f}_i(s) \sum_{k=1}^{n} \hat{\eta}_{ik} \hat{f}_k(s) - f_i(s) \sum_{k=1}^{n} \eta_{ik} f_k(s) \right) ds \right| \\
&\quad - \sum_{i=1}^{n} \left( \sum_{i=1}^{n} u_i^2 F_i \right) \left| \int_0^t \left( \hat{f}_i(s) - f_i(s) \right) ds \right|.
\end{aligned}
\tag{47}
$$

Now, observe that

$$
\begin{aligned}
\sum_{i=1}^{n} &\left| \int_0^t \left( \hat{f}_i(s) \sum_{k=1}^{n} \hat{\eta}_{ik} \hat{f}_k(s) - f_i(s) \sum_{k=1}^{n} \eta_{ik} f_k(s) \right) ds(s) \right| \leq \\
&\leq \int_0^t \sum_{i=1}^{n} \hat{f}_i(s) \sum_{k=1}^{n} \hat{\eta}_{ik} \hat{f}_k(s) \, ds + \int_0^t \sum_{i=1}^{n} f_i(s) \sum_{k=1}^{n} \eta_{ik} f_k(s) \, ds \\
&\leq (\eta + \hat{\eta})t,
\end{aligned}
\tag{48}
$$

and

$$
\sum_{i=1}^{n} \left( \sum_{i=1}^{n} u_i^2 F_i \right) \left| \int_0^t \left( \hat{f}_i(s) - f_i(s) \right) dt \right| \leq \left( \sum_{i=1}^{n} u_i^2 F_i \right) \int_0^t \|\mathbf{f}(t) - \hat{\mathbf{f}}(t)\|_1. \tag{49}
$$

Using these two last relations, inequality (47) may be rewritten in the form

$$\|\mathbf{f}(t) - \hat{\mathbf{f}}(t)\|_1 \geq \Gamma(t) - (\eta + \hat{\eta})t - \left( \sum_{i=1}^{n} u_i^2 F_i \right) \int_0^t \|\mathbf{f}(t) - \hat{\mathbf{f}}(t)\|_1,$$

so that

$$\|\mathbf{f}(t) - \hat{\mathbf{f}}(t)\|_1 \geq \frac{\Gamma(t) - (\eta + \hat{\eta})t}{1 + (\sum_{i=1}^{n} u_i^2 F_i)t},$$

and inequality (45) is achieved.　□

**Remark 12.** *By using inequality* (45), *we see that, if* $\Gamma(t) > (\eta + \hat{\eta})t$, *then an instability appears in the framework* (4). *In addition, it is important to note that this is a condition involving the parameters of the system, i.e., interaction rate and transition probability density.*

*4.3. Numerical Simulations*

This section aims to present some numerical simulations in the framework described by (22). Specifically, the parameters of the system, i.e., interaction rate and transition probability, acquire different values. All the simulations that follow have been performed by using the routine Ode45 of MatLab.

Let $n = 3$, which is three functional subsystems that are taken into account. The initial data are the vector:

$$\mathbf{f}^0 = (3/8, 1/2, 1/8).$$

The interaction rate parameter has the following form:

$$\eta_{hk} = \exp(-\eta\,|h - k|), \qquad \eta > 0.$$

In the first set of simulations, the transition probability is constant, whether the interaction rate varies. Specifically, three cases are considered:

- $\eta = 1$;
- $\eta = 3$;
- $\eta = 6$.

The solution $\mathbf{f}(t) = (f_1(t), f_2(t), f_3(t))$ is of course different from value to value of the interaction rate. Specifically, Figure 1 shows the three plots of the solution $\mathbf{f}(t)$ respectively corresponding to the three different values of $\eta$ listed above. In addition, Figure 2 offers a comparison between the solutions corresponding to the values $\eta = 3$ and $\eta = 3,2$, respectively.

In the second set of simulations, the interaction rate is constant, while the transition probability density acquires different real values. Precisely, Figure 3 shows the three plots of the solution $\mathbf{f}(t)$ respectively corresponding to three different values of $B^i_{hk}$. The considered cases are:

- 
$$B^i_{hk} = c_{ihk}\,\frac{1}{s}\,g(|h - i|), \quad i, h, k \in \{1, 2, \ldots, n\},$$

  where $g$ is a non-increasing function of $|h - i|$ and $s$, and the parameters $c_{ihk}$, for $i, h, k \in \{1, 2, \ldots, n\}$, are positive real numbers, depending on the particular system taken into account;
- $\hat{B}^i_{hk}$ that differs from $B^i_{hk}$ only for $h = 3$ and $k = 2$;
- $B^i_{hk}$ uniform.

Furthermore, Figure 4 shows the solutions corresponding to the values $B^i_{hk}$ and $\hat{B}^i_{hk}$ in the same plot in order to compare their behaviors in time.

It is worth being stressed that the shape of solution strictly depends on the value of the parameters of the system (see Figures 1 and 3) as the results reported in Sections 4.1 and 4.2 show for both the continuous and the discrete framework. Moreover, bearing the Figures 2 and 4 in mind, a small perturbation of a parameter may determine that the related solution has the same shape, but they are not so "close" to each other.

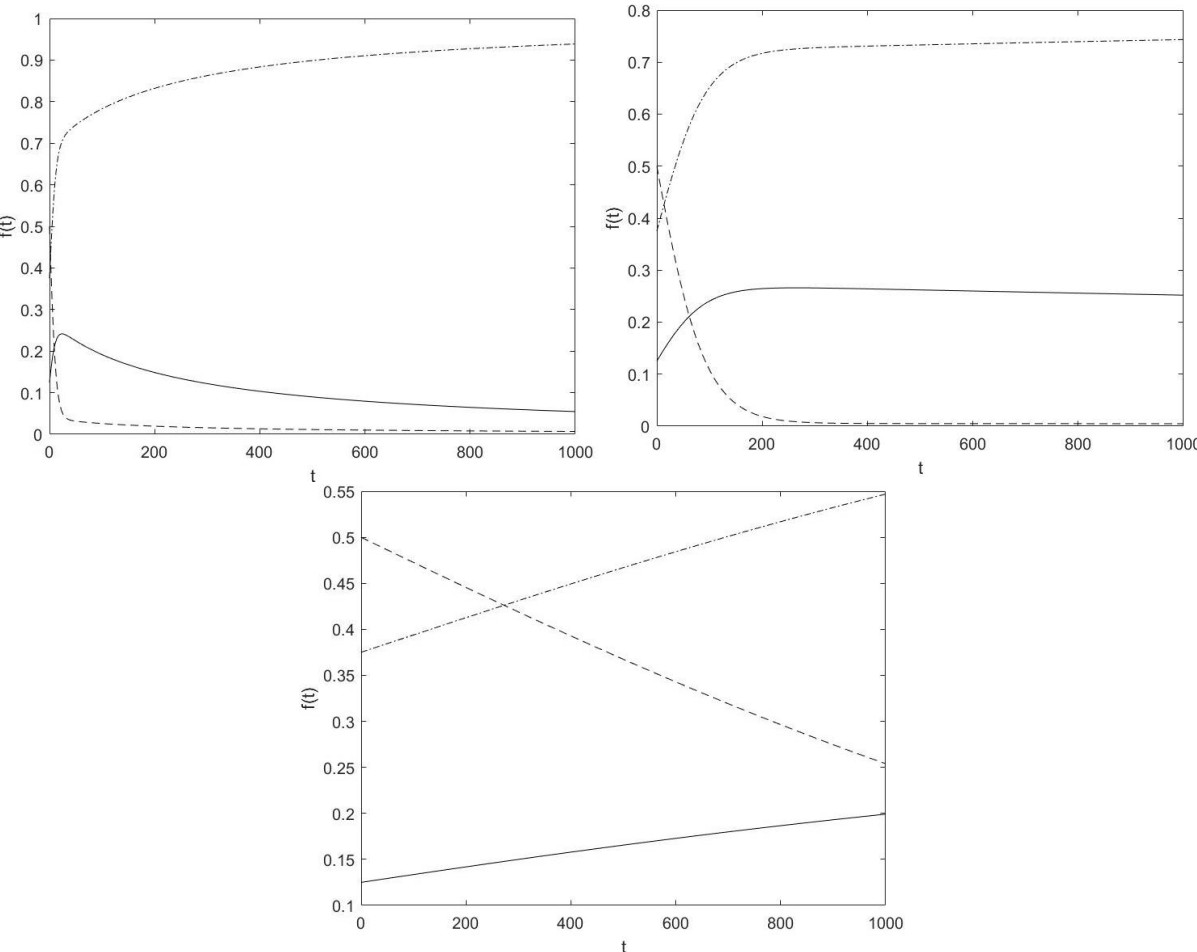

**Figure 1.** From top left to bottom $\eta = 1$, $\eta = 3$, $\gamma = 6$. $f_1$ dot-dashed, $f_2$ dashed, $f_3$ full.

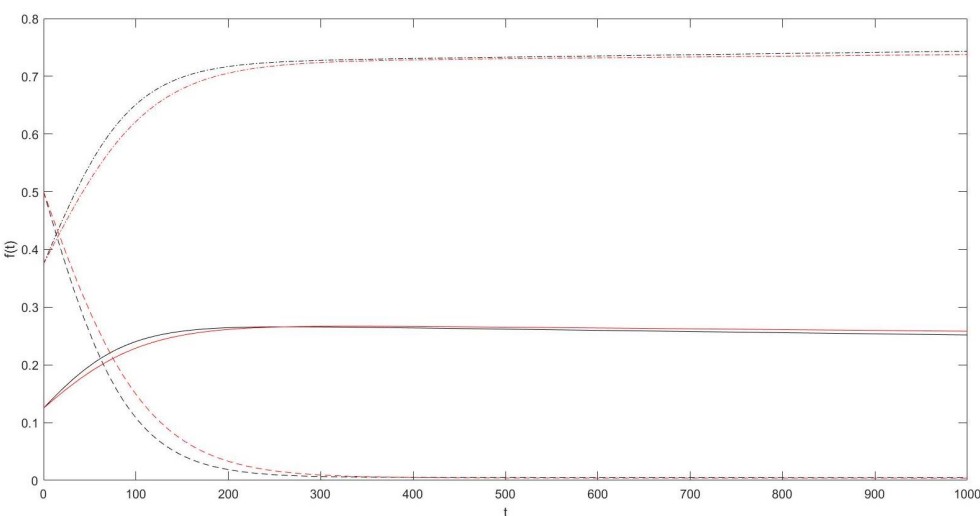

**Figure 2.** In black the solution for $\eta = 3$, in red the solution for $\eta = 3, 2$.

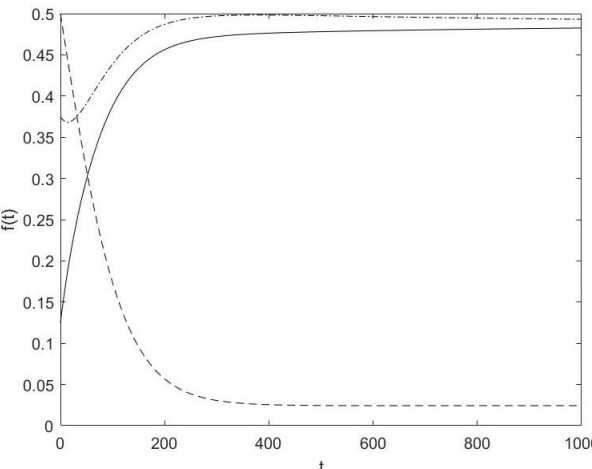

**Figure 3.** From top left to bottom $B_{hk}^i$, $\hat{B}_{hk}^i$ perturbed and uniform distribution. $f_1$ dot-dashed, $f_2$ dashed, $f_3$ full.

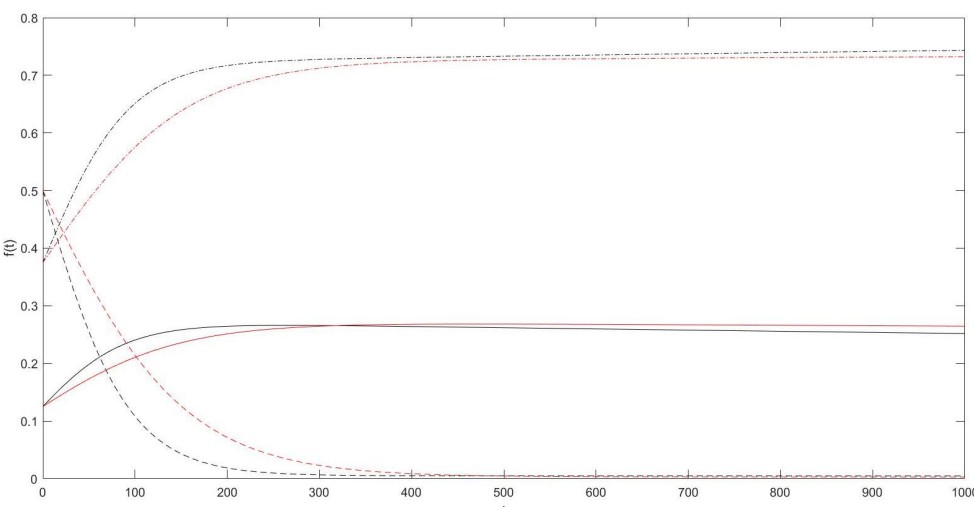

**Figure 4.** In black the solution for $B_{hk}^i$, in red the solution for $\hat{B}_{hk}^i$.

## 5. Conclusions and Research Perspectives

The results proved in Sections 3 and 4 are, in some sense, complementary. The former shows that the difference between two arbitrary solutions of Equation (1), corresponding to two different sets of parameters, i.e., different systems of interaction rates and different probability distributions on the results of interactions, varies with a suitable measure of the difference between the systems of parameters, and the variation is continuous; the latter shows that, if the difference between the interaction rates is sufficiently large in a suitable sense, then the corresponding solutions—though starting from the same initial value—will move at once away from each other and will stay apart at any future time, i.e., their distance has a constant positive lower bound. In addition, the numerical simulations plotted in the figures shown in Section 4.3 seem to give a good visual counterpart of this result.

As far as we are aware, no similar results have been previously reported in the literature about KTAP, perhaps because the study of the dependence of solutions on the perturbations of parameters (interaction rates and transition probabilities) seems to be too difficult in relation to its relevance for applications, so that tackling it is considered as an almost *useless* effort. However, on the contrary, results like the ones found and reported in the present work are probably intended for becoming of the greatest relevance for applications, with special concern with social and economic sciences. In this connection, we can observe that economic interactions in any human society are ruled by the government: in a country in which some commercial transactions are allowed, they will produce exchanges of goods and money, with a subsequent modification of the distribution of wealth; but, in another country, where the same transactions are forbidden, the interaction rate referred to them is zero, and we must expect that the distribution of wealth could not be modified by these transactions, regardless of the values of transition probabilities that are allowed to be the same in both cases. This remark has worked as a suggestion of a search for instability results of the kind of Theorems 3 and 4. Of course, these Theorems cannot be considered as more than a first step on the way towards much more general instability results, for at least the good reason that they only refer to the very special case in which the interaction rates are constant with respect to the couples of states. Accordingly, this research about instability requires to be deepened along at least three lines, which will be the object of future work.

As laid out in the Introduction, KTAP is *not* a theory or simply a model, but a whole scheme of models to describe and—above all—*predict* the behavior of complex systems. In addition, as a matter of fact, our prime scope is its application to human collectivities, in order to suggest some ways to solve the problems raised by many and diffused bad mental habits that control not only human behaviors but also the criteria according to which legislators decide the (inter-)actions that can be allowed and the (inter-)actions that must be forbidden. Laws can modify *both* interaction rates and transition probabilities, so a complete and detailed view of the behaviors they produce could avoid that past mistakes from being repeated in the future.

In this line of thought, first of all, one should find possible conditions of instability in the quite general case in which interaction rates are *arbitrary* functions defined on $D_u \times D_u$: from a purely mathematical viewpoint, this will require in turn a suitable definition of their distance.

Next, one has to find possible instability conditions on the transition probabilities, also in the case in which the interaction rates are left unchanged.

Finally, one has to study the reciprocal influence between the perturbation of interaction rates and the perturbation of transition probabilities. In this connection, it should be noted that Theorem 3 already furnishes a first hint in this direction.

These three lines of search give good and hopefully—on a pragmatic ground—useful perspectives for the development of the study started and reported in the present paper.

**Author Contributions:** Conceptualization, B.C. and M.M.; methodology, B.C. and M.M.; software, B.C. and M.M.; validation, B.C. and M.M.; formal analysis, B.C. and M.M.; writing—original draft preparation, M.M.; writing—review and editing, M.M. All authors have read and agreed to the published version of the manuscript.

**Funding:** This research received no external funding.

**Institutional Review Board Statement:** Not applicable.

**Informed Consent Statement:** Not applicable.

**Data Availability Statement:** Not applicable.

**Acknowledgments:** Marco Menale is supported by the Research Project "ANDROIDS" (AutoNomous DiscoveRy Of depressIve Disorder Signs) for VALERE (VAnviteLli pEr la RicErca), developed by Università degli Studi della Campania "L. Vanvitelli".

**Conflicts of Interest:** The authors declare no conflict of interest.

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
