# Peer review of "Towards the Dependence on Parameters for the Solution of the Thermostatted Kinetic Framework"

_axioms, doi:10.3390/axioms10020059_

Round 1

Reviewer 1 Report

The manuscript deals with the description of complex systems involving a large number of interacting particles exchanging a quantity called activity in the framework of thermostatted kinetic theory.

The continuous dependence of the solution of the kinetic equations on the parameters describing the interaction between particles is demonstrated for continuous  and discrete dependences  of  the distribution functions on particle activity. The different steps of the demontrations are carefully presented and the reading of the proof is pleasant.

I find the proof of the existence of an instability for appropriate parameter values particularly valuable. The results are obtained for constant interaction rates and certainly deserve further investigation.

The analytical results are advantageously illustrated by a numerical study.

The manuscript is basically well written. I however find the abstract too vague and suggest the authors to better sum up their results. I also have a remark about the opposition between deterministic and stochastic features made in lines 30-31 of the introduction. In statistical physics, the resort to a probability distribution is related to the description of a large number of particles but does not always refer to an intrinsically stochastic description, as for example in a master equation associated with Markovian processes.  In particular, the Boltzmann equation is usually considered deterministic in the sense where the trajectory of a particle is described by deterministic rules.  I therefore suggest to revise the introduction regarding the opposition between determinism and stochasticity.

Apart from these minor comments, I recommend the publication of the manuscript in Axioms.

Author Response

Thanks for the suiggestions.

The abstract has been rewritten.

The opposition between determinism and stochasticity has been now addressed in the introduction.

Reviewer 2 Report

This is an interesting, but not well organized and written paper. The main ideas are sufficiently developed, but some integration should be done:

  • The abstract should be expanded by giving more specific results, and the authors should delete some sentences not properly typical of an abstract.
  • The use of non-technical words should be avoided.
  • The quality of the figures should be improved
  • What are the lessons learned from the conclusions other than the observation?
  • The novelty is of this article or is it a validation of a previously established notion/fact?
  • Please rewrite the conclusions to give a better description of the main results. Some philosophical discussions may also be included regarding the findings.

Author Response

Thanks for the suggestions.

Please check the attachment for responses and changes.
